# Effect of NSAIDs Supplementation on the PACAP-, SP- and GAL-Immunoreactive Neurons in the Porcine Jejunum

**DOI:** 10.3390/ijms222111689

**Published:** 2021-10-28

**Authors:** Marta Brzozowska, Barbara Jana, Jarosław Całka

**Affiliations:** 1Department of Clinical Physiology, Faculty of Veterinary Medicine, University of Warmia and Mazury in Olsztyn, Oczapowskiego Str. 13, 10-718 Olsztyn, Poland; calkaj@uwm.edu.pl; 2Institute of Animal Reproduction and Food Research of the Polish Academy of Sciences, Tuwima Str. 10, 10-748 Olsztyn, Poland; b.jana@pan.olsztyn.pl

**Keywords:** NSAIDs, ENS, peripheral nervous system, jejunum, pig

## Abstract

Side effects associated with nonsteroidal anti-inflammatory drugs (NSAIDs) treatment are a serious limitation of their use in anti-inflammatory therapy. The negative effects of taking NSAIDs include abdominal pain, indigestion nausea as well as serious complications such as bleeding and perforation. The enteric nervous system is involved in regulation of gastrointestinal functions through the release of neurotransmitters. The present study was designed to determine, for the first time, the changes in pituitary adenylate cyclase-activating polypeptide (PACAP), substance P (SP) and galanin (GAL) expression in porcine jejunum after long-term treatment with aspirin, indomethacin and naproxen. The study was performed on 16 immature pigs. The animals were randomly divided into four experimental groups: control, aspirin, indomethacin and naproxen. Control animals were given empty gelatin capsules, while animals in the test groups received selected NSAIDs for 28 days. Next, animals from each group were euthanized. Frozen sections were prepared from collected jejunum and subjected to double immunofluorescence staining. NSAIDs supplementation caused a significant increase in the population of PACAP-, SP- and GAL-containing enteric neurons in the porcine jejunum. Our results suggest the participation of the selected neurotransmitters in regulatory processes of the gastrointestinal function and may indicate the direct toxic effect of NSAIDs on the ENS neurons.

## 1. Introduction

The anatomical organization as well as the neurochemical characteristics of intrinsic neural networks responsible for the functioning of intestinal processes appear to be more complex in larger mammals, including humans, than in small laboratory animals [1]. Due to the structural and functional similarities, the porcine intestine is considered a suitable experimental model for studying the physiology of the human enteric nervous system (ENS) as well as its plasticity and response to an external stimulus. The ENS is made up of three intramural ganglionated plexuses: the myenteric plexus (MP), the outer submucous plexus (OSP) and the inner submucous plexus (ISP), neural connections between these plexuses and also with nerve fibers supplying the surrounding tissue (Figure 1) [2]. The intestinal neuronal cells are characterized by wide range of neurotransmitters and/or neuromodulators, the levels of which fluctuates during pathological processes [1,2].

In 1989, pituitary adenylate cyclase-activating polypeptide (PACAP)-38 was isolated for the first time from the ovine hypothalamus, as a peptide with properties to stimulate cAMP production [3]. A year later, Miyata et al. isolated a shorter form peptide with 27 residues corresponding to the N-terminal 27 amino acids of PACAP-38 and amidated C-terminus [4]. PACAP is a part of the vasoactive intestinal peptide (VIP)/secretin/glucagon family [4]. The peptide acts through at least three G protein-coupled VIP/PACAP receptors. The PACAP-specific receptor (PAC1-R) is characterized by lower affinity for VIP than for PACAP, whereas the affinity for VIP and PACAP for the Vasoactive Intestinal Peptide receptors (VPAC1-R and VPAC2-R) is similar [5]. The peptide is available in the intestinal neural structures, where it acts as a neurotransmitter and/or neuromodulator. PACAP has also been disclosed to exert neurotrophic and neuroprotective effects through a combination of anti-apoptotic, anti-inflammatory and antioxidant effects. A neuropeptide is widely distributed in the nervous system as well as in peripheral organs [6].

Substance P (SP), belonging to the tachykinin family, is involved not only in the regulation of gastrointestinal functions, such as motor function and secretion, but also in the development of inflammation. SP exerts its effects through three types of G protein-coupled NK receptors (NK1, NK2 and NK3) [7]. In the wall of intestines, SP is located in the myenteric plexus, submucosal plexus, intrinsic primary afferent neurons (IPANs), as well as in enteroendocrine cells and extrinsic sensory fibers. SP very often coexists with acetylcholine in enteric neurons and fibers and therefore is considered a cholinergic co-mediator [7,8].

Galanin (GAL) was first isolated in 1983 [9]. The presence of GAL has been confirmed both in the central and peripheral nervous system. It has been proven that GAL reveals biological effects by activating specific galanin receptor subtypes: GAL-R1, GAL-R2 and GAL-R3 [9]; indeed, the effect of galanin on the gastrointestinal tract (GIT) is multiple, and GAL inhibits the secretion of gastric acid, as well as active substances such as glucose, insulin and somatostatin. In the digestive system, GAL modifies motility by increasing and decreasing the release of neuroactive substances, and may also act directly by activating receptor/s located at the smooth muscle cells [10].

It is well known that enteric neurons respond to pathological and physiological stimuli such as aging, diet or gastrointestinal disorders by altering the chemical characterization [3,11,12]. In spite of the fact that PACAP, GAL and SP are neuropeptides involved in inflammatory processes, there is still a lack of knowledge regarding the gastrointestinal occurrence of this neurotransmitters during diverse pathological processes, including nonsteroidal anti-inflammatory drugs (NSAIDs)-induced enteropathy. PACAP has widespread expression in the GIT, where it shows protective effects in various intestinal pathologies [6]. Previous research indicated that changes in PACAP distribution in the ENS occurred after administration of low-dose *Salmonella Enteritidis* lipopolysaccharides [13] and during Crohn’s disease [14]. The changes in SP expression within the ENS have been noted in parasite infection [15], carcinoma [16] and inflammatory bowel disease [17], while the changes in the number of GAL-like immunoreactive (LI) enteric nervous structures have been recorded after bisphenol A supplementation [18] and experimentally induced ileal hypertrophy [19].

Nonsteroidal anti-inflammatory drugs are recognized as the most popular over-the-counter medications and accounted for 5% of the prescribed drugs in the world [20]. Besides the anti-inflammatory, analgesic and antipyretic effects, NSAIDs have been reported to be useful in treating pain, fever and inflammation in rheumatic disorders, osteoarthritis and dysmenorrhea. Additionally, aspirin is an effective antiplatelet drug for preventing thrombo-embolic vascular events [20]. However, many multiple placebo-controlled trials and meta-analyses studies show worryingly the adverse effects of NSAIDs as manifested in appearance of symptoms indicating multi-organ system involvement, but gastrointestinal disorders come to the fore [20]. Chatterjee et al. revealed NSAID-related gastrointestinal complications in 30.8% of patients [21]. In turn, Hamid et al. documented the presence of NSAID-related ulcers in 14.7% of patients and special attention should be paid to the fact that duodenal ulcers occurred more frequently than gastric ulcers [22]. Due to a number of systemic complications, it is appropriate to determine the effect of NSAIDs on enteric neurons, not only of the duodenum or stomach, but also other parts of the GIT.

Considering the above, the present study was designed to elucidate, for the first time, the impact of prolonged aspirin, indomethacin and naproxen supplementation (at doses used in the treatment of chronic pain or other chronic condition) on a population of PACAP-, SP- and GAL-LI neurons in the porcine jejunum.

## 2. Results

In the present study, aspirin, indomethacin and naproxen administration caused changes in the neurochemical characteristics of neurons in all parts of the jejunal wall tested. The nature of these fluctuations clearly depended on the type of neurotransmitter studied and the part of the intestinal wall (Figure 2). In treated animals, total number of neurons in each type of the enteric neurons were higher in comparison to control animals. The studies showed statistically significant differences between the number of GAL-LI and SP-LI neurons in the ASA group compared to INDO and NA, as well as between INDO and ASA in all intramural plexuses. In the case of PACAP-positive neurons, the significant differences between ASA and NA in the outer submucosal plexus were observed. A comparison of the PACAP, GAL and SP levels between the intramural plexuses revealed the existence of statistical differences in all experimental groups.

### 2.1. NSAIDs-Induced Changes in Neurochemical Coding of Myenteric Neurons

In animals receiving empty gelatin capsules, 11.87 ± 0.93% PACAP-positive neurons were recorded in the MP (Figure 3A). In turn, slightly more numerous were neurons containing GAL (Figure 3B) and SP (Figure 3C). Treatment with aspirin, indomethacin and naproxen resulted in an increase in the percentage of neurons immunoreactive to all examined neurotransmitters.

In animals treated with aspirin, the highest average percentage increase was observed for GAL-LI neurons by about six percentage points (pp) (Figure 3E). Less visible changes were noticed for PACAP- and SP-positive neurons, where the increase amounted to about 3 pp compared to control (Figure 3D,F).

The indomethacin and naproxen administration also resulted in an increase PACAP, GAL and SP expression. The most visible changes were noted for GAL-IR both in INDO (20.37 ± 0.45%) (Figure 3K) and NA group (23.27 ± 0.94%) (Figure 3H). The number of neuronal cells immunoreactive to PACAP was estimated at 13.73 ± 0.58% in naproxen group (Figure 3G) and 14.66 ± 0.89% in indomethacin group (Figure 3J). In turn, a similar fluctuation of the SP expression degree was observed in indomethacin and naproxen pigs. In these cases, the increase amounted to about 6 pp (Figure 3L) and 4 pp (Figure 3I) in comparison with the control animals respectively.

### 2.2. NSAIDs-Induced Changes in Neurochemical Coding of Outer Submucosal Neurons

In the control animals, the largest populations of enteric neurons located in the OSP showed the presence of PACAP (14.02 ± 1.26%) (Figure 4A). The other substances studied were noted in about 9% of all PGP 9.5-LI cells (Figure 4B,C).

In animals treated with aspirin, a statistically significant increase in the number of PACAP, GAL and SP-positive neurons was noticed in this bowel part. Similar to the MP, ASA in the OSP caused the highest percentage increase for neuronal cells immunoreactive to GAL (17.16 ± 0.6%) (Figure 4E). A less visible increase was observed for neurons immunostained with PACAP (Figure 4D). In contrast, for SP, the increase amounted to about 1 pp in relation to the control animals (Figure 4F).

A similar impact of indomethacin as well as naproxen was noted on the enteric cells located in the OSP. For GAL, a highly significant increase in the number of GAL-IR neurons was found in the NA group (20.56 ± 0.395) (Figure 4H) as well as INDO group (18.91 ± 0.63%) (Figure 4K). Treatment with naproxen resulted in an increase in the number of PACAP- and SP-LI neurons by 2 pp (up to 16.72 ± 1.09% and 16.61 ± 0.87%, respectively) (Figure 4G,I) compared to control pigs. Interestingly, the increase was also observed for SP- (of about 5 pp) (Figure 4L) and PACAP-positive (up to 16.88 ± 0.63%) neurons in INDO group (Figure 4J).

### 2.3. NSAIDs-Induced Changes in Neurochemical Coding of Inner Submucosal Neurons

In the control group, the largest population of inner submucosal neurons showed the PACAP expression. The percentages of PACAP-LI neurons amounted to about 16% of all PGP 9.5-LI neurons (Figure 5A). However, the percentage of perikarya showing the expression of GAL and SP was smaller (14.81 ± 0.73% and 9.24 ± 0.54%, respectively) (Figure 5B,C).

Aspirin supplementation caused an increase in the population of neurons immunoreactive to PACAP (to 18.30 ± 0.46%) (Figure 5D) and GAL (to 21.61 ± 0.68%) (Figure 5E). The slightest visible increase was recorded for SP-positive cells (of about 1 pp) (Figure 5F).

In the INDO group, the most visible increase was found in SP-positive neurons (17.23 ± 0.60%, about 8 pp) (Figure 5L). A less visible fluctuations were revealed for GAL- and PACAP-LI perikarya (18.80 ± 0.62% and 19.04 ± 1.42%, respectively) (Figure 5J and Figure 5K).

The most significant alterations were found among SP-positive (increase of 11 pp) neurons in the naproxen group (Figure 5I). Moreover, the administration of NA resulted in increase in population of PACAP-LI neurons (up to 20.23 ± 0.37%) (Figure 5G). In this group, the increase the number of cells immunostained with GAL was also significant (20.42 ± 0.74%) (Figure 5H).

## 3. Discussion

Serious side effects such as ulcers and perforation as well as enteropathies and mucosal lesions (petechiae, reddened folds, loss of villi) create reduction in the use of NSAIDs. These drugs are thought to demonstrate such effects by the inhibition of cyclooxygenase (COX), resulting in the inhibition of prostaglandin (PG) synthesis at inflamed sites. PG is involved in the regulation of blood flow in the GIT and various mucosal functions such as increasing mucus secretion [20,23,24]. The decrease in PG production is considered to be the main cause of the formation NSAID-induced enteropathy. Although clinical signs of gastroduodenal damage were observed in animals supplemented with aspirin, indomethacin and naproxen, no inflammatory changes in the jejunum were noted. This may suggest that the preclinical effect of treatment with NSAIDs in the jejunum was to alter the neurochemical characteristics of enteric neurons. An increased PACAP, GAL and SP expression in the jejunal enteric neurons, demonstrated in this study, may be a response of the ENS to the harmful effects of NSAIDs administration. Neurons showing the expression of neuronal factors included into the study have been imaged in each submucous (OSP and ISP) and myenteric plexuses in the wall of jejunum. These observations are in line with our previous reports that revealed the changes in neurotransmitter expression in porcine duodenum induced by naproxen and indomethacin [23,25].

The results presented in this study showed that naproxen was the greatest impact on jejunal enteric neurons. In this experimental group, the highest increase in GAL (in MP and OSP) and SP (in OSP and ISP) expression was recorded. In the case of PACAP, studies revealed that naproxen had a more significant effect on ENS neurons than aspirin. It can therefore be speculated that the most significant subclinical effects in the porcine jejunum were caused by naproxen. It is consistent with the observations of macroscopic changes in the stomach and duodenum, where naproxen treatment led to the formation of hyperemia, petechiae, edema, excessive mucus production (duodenum) and ulcers (stomach) [23]. Interestingly, this study also revealed that the highest neuropeptides expression was found in the inner submucosal plexus, which was not unexpected, given the fact that these cells are the most exposed to the damaging effects of NSAIDs.

Changes in the studied neurotransmitters expression in the ENS neurons may result of adaptation to irritants to which the cell is exposed. It is a mechanism of neuronal response to changing and often unfavorable conditions. It is well known that many of the neuroactive substances found in the ENS have neuroprotective properties. One of them is PACAP.

Many authors provide evidence pointing to the protective properties of PACAP [26,27,28]; indeed, the protective effect of PACAP was revealed in a rat model of duodenal ulcer. The intravenous administration of PACAP-27 resulted in an increased production of bicarbonates, which reduced inflammatory changes in the duodenum [29]. PACAP has been proven to play a protective role against autoimmune encephalomyelitis [30]. The importins of the peptide in protection against neurotoxicity, neurodegeneration and spinal cord injury have also been described [31]. Atlasz et al. documented the involvement of PACAP in the protection of retinal neurons against UV-induced damage [32]. Unsurprisingly, the study using the dextran sulfate sodium (DSS)-induced colitis model also confirmed the anti-inflammatory effects of PACAP [33]. It was noted that mice lacking PACAP showed more severe symptoms of colitis. Additionally, aggressive-appearing colorectal cancer was reported in 60% of these animals [34]. Interestingly, in vitro experiments with human colonic tumor cells (HCT-8) have demonstrated the effects of PACAP on proliferation [33]. The authors described PACAP as well as the specific PAC1 receptor in the HCT-8 cell line. Additionally, PACAP-38 has an inhibitory effect on the Fas receptor, suggesting the involvement of PACAP in cell survival [34]. Lelievre et al. described the action of PACAP-27 in human colon adenocarcinoma cell lines. They found that long-term treatment with PACAP or VIP reduced cell proliferation [35].

Galanin, an important neurotransmitter, has also been shown to have neuroprotective effects in both the central and peripheral nervous systems [9]. Meanwhile, GAL has been shown to be involved in the differentiation of mucosal-type mast cell (MMCs) in vivo, while In the same study, the authors suggest that GAL released from the submucosal neurons may contribute to the differentiation and proliferation of MMC during enteritis [36]. Taledo et al. discovered that chronic galanin administration exerts a beneficial effect on the animal model of inflammatory bowel disease (IBD), as evidenced by an improvement in the repair process [37]. According to Matkowskyj et al. the increased myeloperoxidase activity shown in the colon of galanin receptor 1 (GAL1) knockout mice following experimental *Salmonella* infection clearly indicates that GAL is involved in the anti-inflammatory response of innate immunity in the colon [38]. Interestingly, GAL administration reduced the macroscopic damage in the colon mucosa as demonstrated by a rat model of experimentally induced acute colitis [39]. The authors also revealed a reduction in the infiltration degree of polymorphonuclear neutrophils, inhibition of myeloperoxidase activity as well as decreased levels of tumor necrosis factor (TNF)-α and inducible nitric oxide synthase (iNOS) [39]; this corresponds well with the fact that GAL plays an important role in regulating inflammatory processes. Therefore, an increased GAL expression has been recorded in the porcine stomach following hydrochloric acid infusion [40] and acrylamide supplementation [41] and also after treatment with naproxen and indomethacin in porcine duodenum [23,25].

In view of the above, it can be speculated that the increase in the number of GAL-positive neurons is related to neuronal adaptive changes in the ENS in response to the negative effects of NSAIDs. At the same time, it cannot be excluded that treatment with aspirin, naproxen and indomethacin led to damage to the jejunal intramural neurons. This suggests that GAL-negative neurons are more vulnerable to the toxic effects of NSAIDs.

The presented study clearly revealed that the NSAIDs administration induced significant changes in the SP expression in the porcine jejunal neurons. In the digestive tract, substance P modulates inflammation and immune responses [42]. SP, as a sensory neurotransmitter, is involved in the pain transmission. Gastrointestinal disorders, including the side effects of NSAIDs treatment, are often accompanied by pain reactions, which in turn upregulate the expression of sensory neurotransmitters [43]. The data obtained during this study confirm previous reports in which an increase in SP-LI neurons was observed in gastrointestinal diseases, such as IBD [44], during *Helicobacter pylori* infection [45] and under the influence of bisphenol A [18]. SP engages in inflammatory processes mainly by stimulating NK1 receptors on inflammatory cells, as well as by stimulating the production of pro-inflammatory compounds [42]. According to Satheeshkumar and Mohan, SP activates protein kinase C δ (PKC δ) phosphorylation, it synthesizes the nuclear factor-κB (NF-κB) and induces the production of pro-inflammatory cytokines, including TNF-α, IL-1β, IL-8, IL -6 [46]. This fact correlates with the results presented by Azzolina et al. [47]. Authors have been noted that SP administration in an animal model caused mast cell induction, which resulted in increased synthesis of TNF-α and IL-6 [47].

Enteric neurons respond to harmful agents by changing the expression of neurosubstances; often, these changes are noticed before the clinical signs of the pathology become apparent. Therefore, it can be assumed that the changes in the chemical coding of the intramural neurons in the digestive tract are a pre-clinical symptom of the disease; it is also likely that the increase in substance P synthesis, as shown in this study, is related to the onset of activation of the pro-inflammatory cascade caused by NSAIDs treatment.

It is not excluded that changes in the proportions of neurons may be related to the neuronal degeneration of the ENS cells population induced by aspirin, naproxen and indomethacin. It is possible that NSAIDs induce neuronal loss, resulting in an increase in the percentage of SP-, GAL- and PACAP-LI nerve cells. However, to date, there are no literature data describing the effect of NSAIDs on the survival of enteric neurons. Nevertheless, further studies should be conducted to confirm this hypothesis.

## 4. Materials and Methods

### 4.1. Animals

Sixteen cross breed (Pietrain x Duroc) female pigs weighing approx. 20 kg were purchased from a breeding farm in Lubawa (Poland) and were used in the present study. On arrival, the gilts were clinically healthy. Animals were housed in a standard laboratory facility in pens for four animals each, with straw bedding and a light regime of 12 h light/12 h dark. The gilts were given ad libitum access to water and were fed twice a day with commercial compound feed adapted to the needs and age. The handling of animals and all experimental procedures were in accordance with the rules of the National Ethics Commission for Animal Experimentation (Polish Ministry of Science and Higher Education). All experimental protocols were submitted to and approved by the Local Ethics Committee of the University of Warmia and Mazury in Olsztyn (decision no. 54/2017 from 25 July 2017), and we followed guidance in EU Directive 2010/63/EU for animal experiments.

### 4.2. Experimental Procedures

After an adaptation period of 7 days, the animals were weighed and randomly divided into four experimental groups: control (C, *n* = 4), aspirin (ASA, *n* = 4), indomethacin (INDO, *n* = 4) and naproxen (NA, *n* = 4), with four gilts each. The control animals received empty gelatin capsules, while the pigs in the studies groups were given acetylsalicylic acid (ASA group: Aspirin; Bayer Bitterfeld GmbH; Bitterfeld-Wolfen, Germany; 100 mg/kg b. w.; per os), indomethacin (INDO group-Metindol Retard; PharmaSwiss Česká republika s.r.o., Republika Czeska; 10 mg/kg b. w.; per os) and naproxen (NA group-Apo-Napro; Apotex Europe B.V.; Netherlands; 50 mg/kg b. w.; per os). Gilts were weighed once a week to determine the exact doses of medication administered.

The experimental part of the study lasted 28 days. After this time, the animals were pre-treated with azaperone (Stresnil; Janssen Pharmaceutica N.V., Belgium, 2 mg/kg b. w., i.m.). Then, 30 min later, the pigs were given the main anesthetic. Animals were euthanized by intravenous infusion with sodium thiopental (Thiopental, Sandoz, Kundl-Rakúsko, Austria). Immediately after total loss of vital functions, fragments of the jejunum (located about 40 cm after gastric pylorus) were collected from all pigs. In the next stage of the experiment, the material was fixed by immersion in a solution of 4% buffered paraformaldehyde (pH 7.4) for 1 h. The tissues were then washed three times in 0.1 M phosphate buffer (pH 7.4) with a 24 h buffer exchange. Subsequently, the material was transferred into 18% phosphate buffered sucrose and stored at 4 °C for at least 3 weeks. After this time, the collected sections of the jejunum were embedded in Tissue-Tek OCT, cut into 14 µm thick sections using a cryostat (Microm, type HM525, Walldorf, Germany) and fixed on degreased microscope slides.

### 4.3. Immunofluorescence Procedures

The jejunal sections were subjected to double fluorescence immunohistochemical staining to determine the phenotype of enteric neurons in experimental and control animals; these methods were the same as described in a previous paper [48]. In the first stage of the study, the slides were dried at room temperature for one hour and then washed in 0.1 M phosphate buffered saline (PBS, pH 7.4) for 3 × 10 min. The material was placed in a humidified chamber and incubated in a “blocking mixture” composed of 10% horse serum and 0.1% bovine serum albumin in 0.1 M PBS, 1% Triton X-100, 0.05% Thimerosal and 0.01% sodium azide (for 1 h in room conditions), rinsed three times in PBS and incubated overnight with primary antisera against protein gene-product 9.5, used here as a pan neuronal marker and against one of the other neuronal substances studied, that is substance P (SP), galanin (GAL) or is pituitary adenylate cyclase-activating peptide (PACAP). The following day, the slices of jejunum were incubated (humid chamber, 1 h, room conditions) with polyclonal donkey anti-rabbit, anti-mouse and anti-guinea pig IgG antibodies: Alexa Fluor 488 or 546 for one hour to visualize the complexes “antigen-primary antibody”. Primary and donkey IgG antibodies, work dilutions and supplier are presented in Table 1. After washing the tissues again with PBS (3 × 10 min), the slides were covered with polyethylene glycol/glycerin solution. The typical specificity test controls (standard controls, i.e., omission, pre-absorption for the neuropeptide antisera with appropriate antigen as well as replacement of primary antisera by non-immune sera) were performed to verify specificity of immunohistochemical labeling.

### 4.4. Counting and Statistics

To determine the changes in SP, GAL and PACAP expression in jejunal enteric neurons, the tissues were analyzed using an Olympus BX51 microscope (Olympus, Hamburg, Germany) equipped with epi-illumination fluorescence filters. Micrographs were taken using a digital monochromatic camera (Olympus XM 10). The microscope was also equipped with cellSens Dimension Image Processing software, with which photographic documentation was prepared as well as the superimposition of two fluorescence channels. To determine the percentage of PACAP-positive (analogously GAL- and SP-positive) neurons in the jejunum from each type of plexus (MP, OSP and ISP), the number of immunoreactive neurons relative to the tested neurotransmitter was expressed as a percentage of the total number of PGP 9.5-positive cells. A total number of 500 PGP 9.5 neurons were counted. In this case, PGP 9.5-positive cells were considered as 100%, while only neurons with a clearly visible cell nucleus were analyzed in the study. To prevent double counting of neurons, the sections were located at least 100 µm apart. After careful analysis, the obtained data were also pooled and presented as the mean ± standard deviation (SD). The statistical analysis was performed with one-way analysis of variance (ANOVA) with Duncan’s multiple-range test using Statistica 13 software (StatSoft Inc, Tulsa, OK, USA). A *p*-value of 0.05 or less was declared statistically significant.

## 5. Conclusions

Long-term administration of aspirin, indomethacin and naproxen induced a significant increase in GAL-, SP- and PACAP-LI neurons in the porcine jejunum. The observed changes may be caused by the direct influence of NSAIDs on enteric neurons or the inflammation accompanying the treatment. The registered increase in expression of selected neurotransmitters may be related to the participation of neuropeptides in the regulatory processes of the gastrointestinal inflammation. As the negative effects of taking NSAIDs cause damage to the digestive system, the participation of PACAP, GAL and SP in the immune response may be used in the future as a tool in the treatment of digestive disorders.

## Figures and Tables

**Figure 1 ijms-22-11689-f001:**
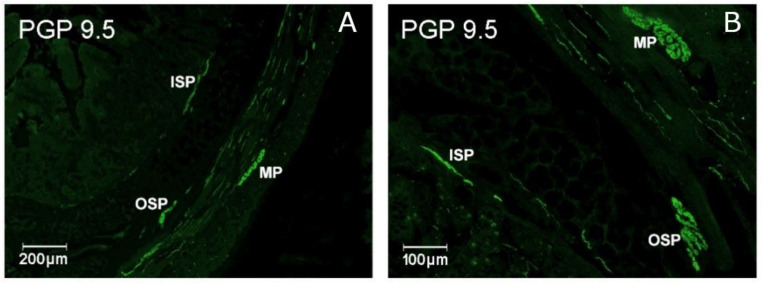
Organization of the enteric nervous system in the porcine jejunum demonstrated by the labeling with protein gene-product 9.5 (PGP 9.5)—used as a pan neuronal marker. Elements of the enteric nervous system: MP, myenteric plexus; OSP, intestinal outer submucosal plexus; and ISP, intestinal inner submucosal plexus. Scale bars indicate 200 µm (**A**) and 100 µm (**B**).

**Figure 2 ijms-22-11689-f002:**
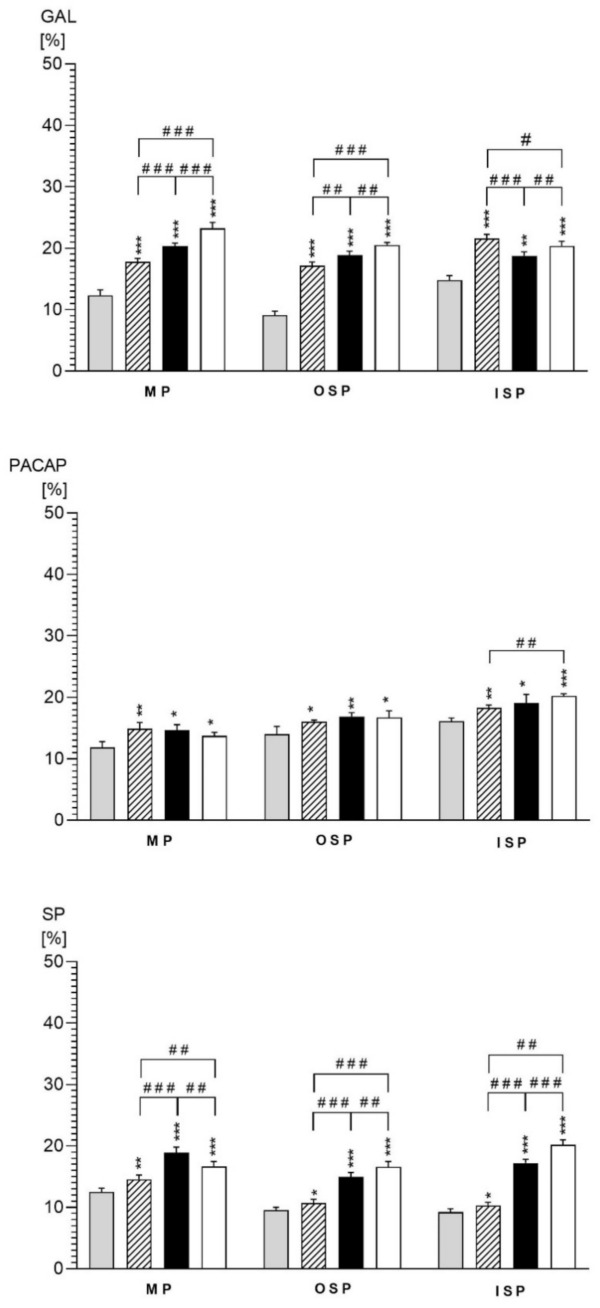
Percentages of PACAP-, GAL- and SP-LI perikarya in the wall of the jejunum in the control (grey bars), aspirin (hatched bars), indomethacin (black bars) and naproxen (white bars) animals. * *p* < 0.05, ** *p* < 0.01, *** *p* < 0.001—indicate differences between the C group and experimental groups for the same neuronal populations; # *p* < 0.05, ## *p* < 0.01, ### *p* < 0.001—indicate differences between ASA vs. INDO vs. NA for population of GAL-, PACAP- and SP-positive neurons in MP, OSP and ISP.

**Figure 3 ijms-22-11689-f003:**
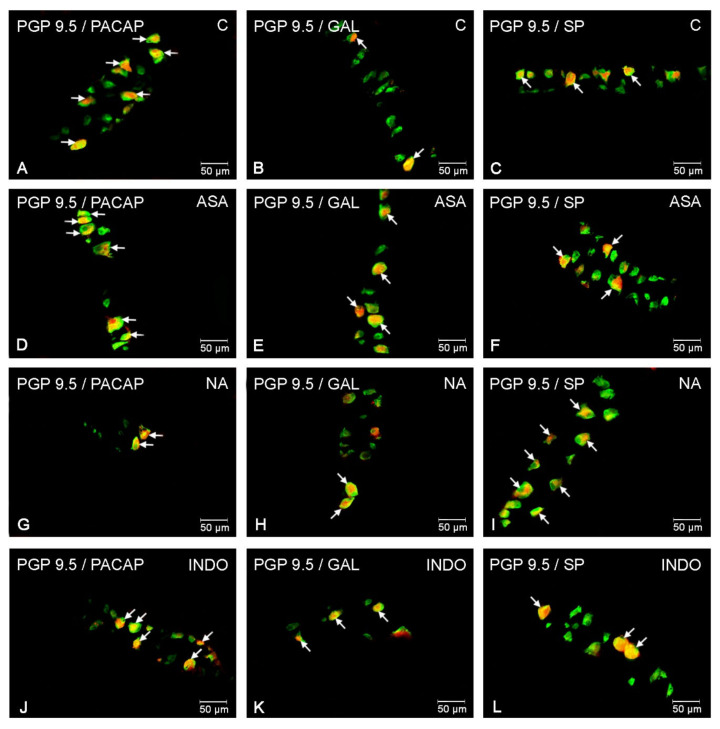
Myenteric neurons of porcine jejunum containing the protein gene-product 9.5 (PGP 9.5) (used as pan neuronal marker) and PACAP, GAL and SP in physiological conditions (**A**-**C**) and following aspirin (**D**–**F**), naproxen (**G**–**I**), indomethacin (**J**–**L**) treatment. The photographs were created by digital superimposition of two color channels (green for PGP 9.5 and red for selected neurotransmitter). The arrows in the photos point to co-localization of both antigens in the studied cells.

**Figure 4 ijms-22-11689-f004:**
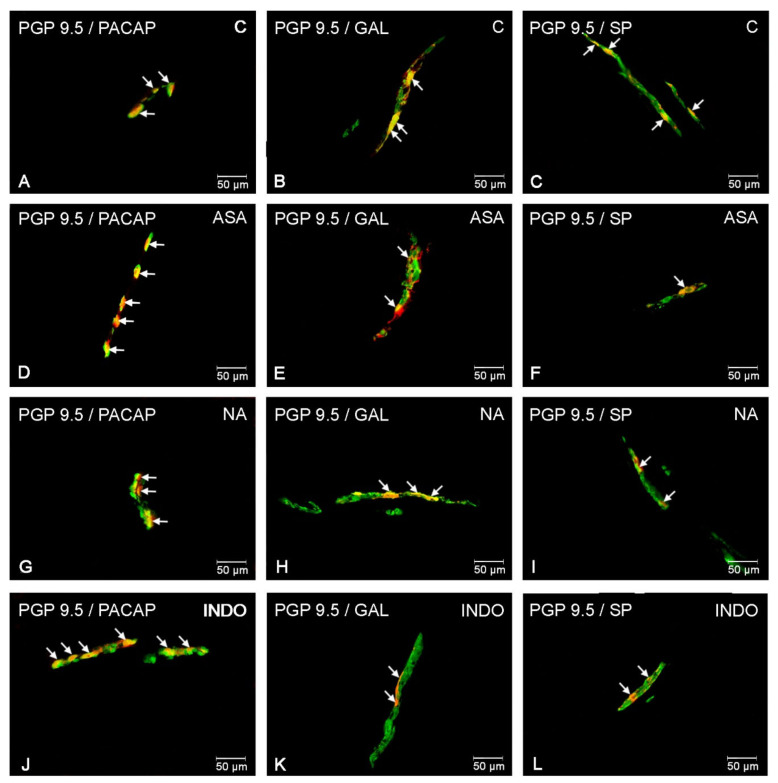
Outer submucosal neurons of porcine jejunum containing the protein gene-product 9.5 (PGP 9.5) (used as a pan neuronal marker) and PACAP, GAL and SP in physiological conditions (**A**-**C**) and following aspirin (**D**–**F**), naproxen (**G**–**I**), indomethacin (**J**–**L**) treatment. The photographs were created by digital superimposition of two color channels (green for PGP 9.5 and red for selected neurotransmitter). The arrows in the photos point to co-localization of both antigens in the studied cells.

**Figure 5 ijms-22-11689-f005:**
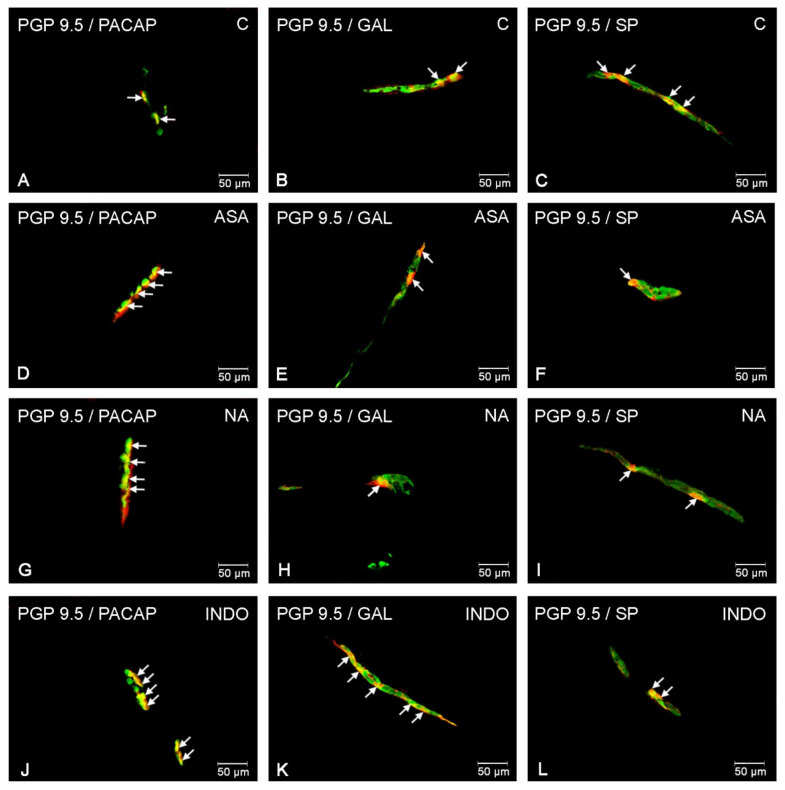
Inner submucosal neurons of porcine jejunum containing the protein gene-product 9.5 (PGP 9.5) (used as a pan neuronal marker) and PACAP, GAL and SP in physiological conditions (**A**–**C**) and following aspirin (**D**–**F**), naproxen (**G**–**I**), indomethacin (**J**–**L**) treatment. The photographs were created by digital superimposition of two color channels (green for PGP 9.5 and red for selected neurotransmitter). The arrows in the photos point to co-localization of both antigens in the studied cells.

**Table 1 ijms-22-11689-t001:** Reagents used in immunofluorescence method.

Primary Antibodies				
Antigen	Code	Dilution	Species	Supplier
PGP 9.5	7863-2004	1:1000	Mouse	BioRad, Hercules, CA, USA
SP	8450-0004	1:1000	Rabbit	BioRad, Hercules, CA, USA
GAL	4600-5004	1:2000	Rabbit	Biogenesis, UK
PACAP-27	T-5039	1:3000	Guinea Pig	Peninsula, San Carlos, CA, USA
**Secondary antibodies**				
**Reagents**		**Dilution**	**Supplier**	
Alexa fluor 488 donkey anti-mouse IgG		1:1000	ThermoFisher Scientific, Waltham, MA, USA	
Alexa fluor 546 donkey anti-rabbit IgG		1:1000	ThermoFisher Scientific, Waltham, MA, USA	
Alexa fluor 546 donkey anti-guinea pig IgG		1:1000	ThermoFisher Scientific, Waltham, MA, USA	

## Data Availability

Data is contained within the article.

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
