# Peer review of "Effect of NSAIDs Supplementation on the PACAP-, SP- and GAL-Immunoreactive Neurons in the Porcine Jejunum"

_ijms, 2021, doi:10.3390/ijms222111689_

Round 1

Reviewer 1 Report

The manuscript of Brzozowska et al. it is an interesting work on the effects of three NSAIDs on the expression of PACAP, SP and GAL in the porcine jejunum. The authors conclude from their results that the variation of expression of the molecules considered can be correlated to some of the side effects of NSAIDS on the gastrointestinal system.

The work is well written but needs revisions.

The biggest problem I think is in the images and in the processing of the results obtained.

The three images of immunofluorescence present only the digital superimpositions of the antibodies tested, although in the three captions you read: "The right column of the photographs were created by digital superimposition of two color channels (green for pgp 9.5 and red for selected neurotransmitter)"

There are no photos of the individual fluorescence channels that instead it is necessary to insert.

Furthermore, the arrows do not always indicate overlapping areas, as can be seen clearly in figures 1A, 1F, 2C, 2D, 2J, where some arrows indicate red nuclei. The authors in the Materials and Metodhs say they used the Olympus cellSens Dimension Image Processing : did the overlapping immunofluorescence analysis with this software? You need to specify it.

Moreover, it would also need to insert images to the light microscope  of the tissue on which immunofluorescences were made

Pag.9 line 311: “secondary antibodies linked to various to visualize….” Various fluorochromes? Something is missing

Author Response

Reviewer 1

  1. The manuscript of Brzozowska et al. it is an interesting work on the effects of three NSAIDs on the expression of PACAP, SP and GAL in the porcine jejunum. The authors conclude from their results that the variation of expression of the molecules considered can be correlated to some of the side effects of NSAIDS on the gastrointestinal system.

The work is well written but needs revisions.

The biggest problem I think is in the images and in the processing of the results obtained.

Authors answer:

Thank you for your valuable comments.

  1. The three images of immunofluorescence present only the digital superimpositions of the antibodies tested, although in the three captions you read: "The right column of the photographs were created by digital superimposition of two color channels (green for pgp 9.5 and red for selected neurotransmitter)"

Furthermore, the arrows do not always indicate overlapping areas, as can be seen clearly in figures 1A, 1F, 2C, 2D, 2J, where some arrows indicate red nuclei. The authors in the Materials and Metodhs say they used the Olympus cellSens Dimension Image Processing : did the overlapping immunofluorescence analysis with this software? You need to specify it.

Authors answer:

Thank you very much for your valuable comments. The figures included in the manuscript demonstrate representative photographs showing PACAP-, GAL-, and SP-LI neurons in animals of each experimental group. These photos were made by superimposing two photos with the Olympus cellSens Dimension Image Processing.

As the reviewer rightly noted, the arrows do not always indicate the co-localization of PGP 9.5 and PACAP/GAL/SP. The authors corrected the photos and the descriptions of the figures. (see Figures 3-5)

  1. There are no photos of the individual fluorescence channels that instead it is necessary to insert.

Authors answer:

The reviewer rightly points out that photos from individual fluorescence channels are more attractive. However, the authors were keen on showing PACAP, GAL and SP positive cells from all study groups as well as from each enteric plexuses (MP, OSP, ISP). Presenting all the photos from each channels would create as many as 12 figures. Figures with superimposed fluorescence channels are often included in publications relating to the study of the enteric nervous system (e.g. doi: 10.3390/ani10040555 and doi: 10.3390/ijms221910308).

However, if the reviewer indicates that creating such a number of photos is appropriate and enriches the manuscript, the authors will prepare the figures according to the recommendations.

  1. Moreover, it would also need to insert images to the light microscope of the tissue on which immunofluorescences were made

Authors answer:

The authors enriched the manuscript with photos from a fluorescence microscope showing the structure of the collected tissues (see Figure 1). Unfortunately, due to the end of the experimental studies, it is not possible to take photos from a light microscope.

  1. 9 line 311: “secondary antibodies linked to various to visualize….” Various fluorochromes? Something is missing

Authors answer:

As suggested by the reviewer, additional data describing the secondary antibodies have been added (see line 348-352).

Reviewer 2 Report

Overall

In the manuscript entitled ‘Effect of NSAIDs Supplementation on the PACAP-, SP- and GAL-immunoreactive Neurons in the Porcine Jejunum’, Brzozowska and coworkers demonstrated the impact of prolonged aspirin, indomethacin and naproxen supplementation on a population of PACAP-, SP- and GAL-LI neurons in the porcine jejunum for the first time. NSAIDs are the most widely prescribed drugs in the world. NSAIDs are responsible for ~30% of hospital admissions for adverse drug reactions, including GI complications like enteropathy, ulcers and GI bleeding. The pathophysiological background is unclear in case of enteropathies, especially uncommon sites like jejunum. In my opinion, the present finding is an important addition to this field and could give a good basis for future studies. The title accurately reflects the content of the manuscript, the abstract is complete and summarizes the most important information. The introduction section gives an insight into the previous findings of the field. However, it seems to be slightly surface-level. The experiment is well-designed and the methodology is accurate. The manuscript would be of greater interest if further statistical analyses were performed, like comparing the effects of the three NSAIDs. The results are described concisely. The conclusions are mostly supported by the results, authors summarize the novel findings, their relation to previous findings and their impact on future studies in a straightforward way. The figures of the paper are clear and high in quality.

Strengths: the good experimental model and the high quality of double fluorescence immunohistochemical staining.

Limitations: the introduction and discussion seems to be slightly surface-level in certain points. The authors should expand some parts.

I have the following comments:

Introduction

The authors should expand the introduction part regarding PACAP:

1) PACAP has two biologically active forms: PACAP-27 and PACAP-38. PACAP-38 was discovered in 1989 (10.1016/0006-291X(89)91757-9) and PACAP-27 was isolated a year later (10.1016/0006-291x(90)92140-u). PACAP-38 is the dominant form in mammals. I would propose to cite this two original article instead of ref: 3 and 4.

2) Line 41: “…to stimulate cAMP accumulation…” I think cAMP production fits better to this sentence.

3) In case of SP and GAL the authors described their receptors (NK1,NK2, NK3 and GAL-R1, GAL-R2, GAL-R3). In case of PACAP it was written: “the peptides act through at least three G protein-coupled VIP/PACAP receptors”. This part should be expanded as PACAP and VIP share two common G protein-coupled receptors: VPAC1-R and VPAC2-R, while PACAP also has an additional specific receptor: PAC1-R. (10.1124/pr.109.001370)

4) Line 42. “The peptides” … the plural form is not necessary or use PACAP instead of this.

5) I would suggest deepening the introduction section more, regarding the general effects of PACAP.  Early studies already pointed out the robust neurotropic and neuroprotective effects of PACAP in vitro and in vivo through a combination of anti-apoptotic, anti-inflammatory and antioxidant effects. Numerous studies have provided proof that PACAP exerts trophic and protective effects not only in the nervous system but also in several peripheral cell types and organs, like GI tract. A recent review summarized the protective effects of PACAP in peripheral organs, including the digestive system (10.3389/fendo.2020.00377). This review could also useful to expand the lines (between 70-72) dealing with the changes of PACAP levels under pathological GI processes or to expand the discussion section.

6) “NSAIDs have been reported to be useful in critical disorders such as cancer and heart attack” This sentence seems to be too “general”. NSAIDs are recommended for mild pain and are the first step in treating chronic pain, including the treatment of cancer pain or chronic musculo-skeletal pain. They are also commonly prescribed in the setting of acute pain. Only a few NSAIDs are used as a part of primary/secondary prevention in case of cardiovascular, cerebrovascular diseases or in some case of arrhythmia because their antiplatelet effect. It’s a big difference clinically and pharmacologically. The antiplatelet role of aspirin could be observed in case of low dose administration via acting mainly on the constitutive form of COX-1. Indomethacin and naproxen don’t have this antiplatelet role.  The authors should separate the clinical indications and describe the main clinical applications of aspirin (low and normal doses), indomethacin and naproxen to highlight the translational potential this research. Regarding to aspirin the authors should state, whether the aspirin dose in this study was a prolonged low dose aspirin therapy (like in case of antiplatelet therapy) or it was a prolonged normal dose therapy (like in case of the treatment of chronic pain or other chronic condition).

Results

7) Table 1 summaries the results of the whole experiment. In my opinion, it would be more informative if this table were replaced by a graph, the same way as the author did in their previous article (Reference 25: Fig.1)  

8) Line 126 and 150: this sections have the same number (2.2.)

9) The changes in GAL-, PACAP-, SP-LI was discussed in case of MP, OSP and ISP for each medicine group under section 2.1., 2.2. and 2.3. There is no data about the possible differences between the ASA vs. INDO vs. NA in MP, OSP and ISP. The manuscript would be of greater interest if changes between ASA, INDO and NA were compered too. It would be also interesting to examine, is there any difference in cases, when the PACAP-, GAL- and SP-LI levels are compared between the anatomical layers (f.e., changes in PACAP-LI in MP vs. OSP vs. ISP).

Discussion

10) “The gastrointestinal mucosal injury in the treatment of chronic inflammation by NSAIDs creates reduction in the use of these drugs” The authors should rephrase and expand this sentence to highlight that mucosal injury is a common and harmful side-effect of NSAID treatment. It would be also useful to highlight the differences between enteropathy and types of GI injuries (mucosal injury refers to erosion, while ulcers are deeper, so it’s not “just” a mucosal injury).

11) What does this mean: “no clinical signs of gastrointestinal injury were observed”? Is this macroscopic observation true in case of jejunum or was the whole GI tract examined? Was there any visible change in the typical places of side-effects (stomach, duodenum)?

12) With the lack of clinical signs and classic morphological changes, how do we know that the prolonged NSAIDs therapy in this experiment resulted in dysfunction of enteric neurons? Can we hypothesize that there is a dysfunction based only on the change of expression of selected neurotransmitters?

13) The lines between 195-208 deal with the role of PACAP in immune functions and its cytoprotective effects. In this section the authors should focus more on the protective effects of PACAP in GI tract than the immune role. It would improve the quality of the article, if the authors could highlight the importance of PACAP in human conditions, because it’s therapeutic and biomarker potential have also been investigated in various human pathological conditions with encouraging results.

14) The manuscript would be of greater interest if the possible differences between the ASA vs. INDO vs. NA in MP, OSP and ISP should be also discussed as the changes in PACAP/SP/GAL-LI in MP vs. OSP vs. ISP.

15) Lines: 259-260. The authors hypothesize, that NSAIDs might induce neuronal loss, resulting in an increase in the percentage of PACAP-, SP- and GAL-LI nerve cells. Was the presumed neuronal loss examined in this experiment?

Materials and methods

16) This part of article would benefit from subsectioning f.e., animals, preparation of biological samples, IF method, statistical analyses.

17) Table 2 shows the reagents used in IF method. The PACAP antibody used in this experiment was PACAP1-27 antibody based on its code. The 38 amino acid residue (PACAP1-38, or PACAP38) is the predominant form in mammals, while PACAP27 represents only the ~10 % of total PACAP in the body. Why did the authors choose PACAP1-27 antibody instead of PACAP1-38?

Author Response

You will find included corrected version of our manuscript entitled “Effect of NSAIDs Supplementation on the PACAP-, SP- and GAL-Immunoreactive Neurons in the Porcine Jejunum” – Marta Brzozowska, Barbara Jana and JarosÅ‚aw CaÅ‚ka

We appreciate the thorough review. All text improvements of our manuscript have been done in red font.

Here are correction and comments from the editors and reviewers:

__________________________________________________________________________________

Reviewer 2

  1. In the manuscript entitled ‘Effect of NSAIDs Supplementation on the PACAP-, SP- and GAL-immunoreactive Neurons in the Porcine Jejunum’, Brzozowska and coworkers demonstrated the impact of prolonged aspirin, indomethacin and naproxen supplementation on a population of PACAP-, SP- and GAL-LI neurons in the porcine jejunum for the first time. NSAIDs are the most widely prescribed drugs in the world. NSAIDs are responsible for ~30% of hospital admissions for adverse drug reactions, including GI complications like enteropathy, ulcers and GI bleeding. The pathophysiological background is unclear in case of enteropathies, especially uncommon sites like jejunum. In my opinion, the present finding is an important addition to this field and could give a good basis for future studies. The title accurately reflects the content of the manuscript, the abstract is complete and summarizes the most important information. The introduction section gives an insight into the previous findings of the field. However, it seems to be slightly surface-level. The experiment is well-designed and the methodology is accurate. The manuscript would be of greater interest if further statistical analyses were performed, like comparing the effects of the three NSAIDs. The results are described concisely. The conclusions are mostly supported by the results, authors summarize the novel findings, their relation to previous findings and their impact on future studies in a straightforward way. The figures of the paper are clear and high in quality.

Strengths: the good experimental model and the high quality of double fluorescence immunohistochemical staining.

Limitations: the introduction and discussion seems to be slightly surface-level in certain points. The authors should expand some parts.

Authors answer:

Thank you for your valuable comments.

INTRODUCTION

  1. PACAP has two biologically active forms: PACAP-27 and PACAP-38. PACAP-38 was discovered in 1989 (10.1016/0006-291X(89)91757-9) and PACAP-27 was isolated a year later (10.1016/0006-291x(90)92140-u). PACAP-38 is the dominant form in mammals. I would propose to cite this two original article instead of ref: 3 and 4.

Authors answer:

Thank you for your valuable comments. Recommended changes have been included in the manuscript (lines 45-48).

  1. Line 41: “…to stimulate cAMP accumulation…” I think cAMP production fits better to this sentence.

Authors answer:

Thank you for the comments. According to the reviewer's recommendations, the word has been changed (line 46).

  1. In case of SP and GAL the authors described their receptors (NK1,NK2, NK3 and GAL-R1, GAL-R2, GAL-R3). In case of PACAP it was written: “the peptides act through at least three G protein-coupled VIP/PACAP receptors”. This part should be expanded as PACAP and VIP share two common G protein-coupled receptors: VPAC1-R and VPAC2-R, while PACAP also has an additional specific receptor: PAC1-R. (10.1124/pr.109.001370)

Authors answer:

Thank you for the comment. As recommended, the required information has been added (lines 49-52).

  1. Line 42. “The peptides” … the plural form is not necessary or use PACAP instead of this.

Authors answer:

Thank you for the comments. In line with the recommendations, the word has been changed (line 49).

  1. I would suggest deepening the introduction section more, regarding the general effects of PACAP. Early studies already pointed out the robust neurotropic and neuroprotective effects of PACAP in vitro and in vivo through a combination of anti-apoptotic, anti-inflammatory and antioxidant effects. Numerous studies have provided proof that PACAP exerts trophic and protective effects not only in the nervous system but also in several peripheral cell types and organs, like GI tract. A recent review summarized the protective effects of PACAP in peripheral organs, including the digestive system (10.3389/fendo.2020.00377). This review could also useful to expand the lines (between 70-72) dealing with the changes of PACAP levels under pathological GI processes or to expand the discussion section.

Author answer:

The authors thank you for your valuable comment. As suggested, the introduction to PACAP has been changed. Necessary information has been added (lines 54-57 and 79-81).

  1. “NSAIDs have been reported to be useful in critical disorders such as cancer and heart attack” This sentence seems to be too “general”. NSAIDs are recommended for mild pain and are the first step in treating chronic pain, including the treatment of cancer pain or chronic musculo-skeletal pain. They are also commonly prescribed in the setting of acute pain. Only a few NSAIDs are used as a part of primary/secondary prevention in case of cardiovascular, cerebrovascular diseases or in some case of arrhythmia because their antiplatelet effect. It’s a big difference clinically and pharmacologically. The antiplatelet role of aspirin could be observed in case of low dose administration via acting mainly on the constitutive form of COX-1. Indomethacin and naproxen don’t have this antiplatelet role. The authors should separate the clinical indications and describe the main clinical applications of aspirin (low and normal doses), indomethacin and naproxen to highlight the translational potential this research. Regarding to aspirin the authors should state, whether the aspirin dose in this study was a prolonged low dose aspirin therapy (like in case of antiplatelet therapy) or it was a prolonged normal dose therapy (like in case of the treatment of chronic pain or other chronic condition).

Author answer:

The authors thank you for your valuable comment. Necessary information has been added (lines 89-93 and 103-104).

RESULTS

  1. Table 1 summaries the results of the whole experiment. In my opinion, it would be more informative if this table were replaced by a graph, the same way as the author did in their previous article (Reference 25: Fig.1)

Author answer:

Thank you for your comment. The results are presented in the form of a graph (see Figure 2).

  1. Line 126 and 150: this sections have the same number (2.2.)

Author answer:

Thank you for your valuable information. Recommended changes have been included in the manuscript.

  1. The changes in GAL-, PACAP-, SP-LI was discussed in case of MP, OSP and ISP for each medicine group under section 2.1., 2.2. and 2.3. There is no data about the possible differences between the ASA vs. INDO vs. NA in MP, OSP and ISP. The manuscript would be of greater interest if changes between ASA, INDO and NA were compered too. It would be also interesting to examine, is there any difference in cases, when the PACAP-, GAL- and SP-LI levels are compared between the anatomical layers (f.e., changes in PACAP-LI in MP vs. OSP vs. ISP).

Author answer:

Thank you for the comment. As recommended, the authors added information on statistical significance between ASA vs. INDO vs. NA in all jejunal plexuses. The significance in PACAP, SP and GAL levels in all experimental groups between individual enteric plexuses was also defined (Figure 2).

DISCUSSION

  1. “The gastrointestinal mucosal injury in the treatment of chronic inflammation by NSAIDs creates reduction in the use of these drugs” The authors should rephrase and expand this sentence to highlight that mucosal injury is a common and harmful side-effect of NSAID treatment. It would be also useful to highlight the differences between enteropathy and types of GI injuries (mucosal injury refers to erosion, while ulcers are deeper, so it’s not “just” a mucosal injury).

Author answer:

Thank you for your valuable comment. The authors added the necessary information (line 193-195).

  1. What does this mean: “no clinical signs of gastrointestinal injury were observed”? Is this macroscopic observation true in case of jejunum or was the whole GI tract examined? Was there any visible change in the typical places of side-effects (stomach, duodenum)?

Author answer:

Thank you for your valuable comment. This sentence applies to the jejunum, where no signs of inflammation have been reported. Macroscopic changes were seen in the duodenum and stomach of animals treated with NSAIDs. Relevant information has been added (line 199-203).

  1. With the lack of clinical signs and classic morphological changes, how do we know that the prolonged NSAIDs therapy in this experiment resulted in dysfunction of enteric neurons? Can we hypothesize that there is a dysfunction based only on the change of expression of selected neurotransmitters?

Author answer:

Thank you for your valuable suggestions. Authors agree with the reviewer. The use of the word "dysfunction" is completely inappropriate. Recommended changes have been included in the manuscript (line 202-203).

  1. The lines between 195-208 deal with the role of PACAP in immune functions and its cytoprotective effects. In this section the authors should focus more on the protective effects of PACAP in GI tract than the immune role. It would improve the quality of the article, if the authors could highlight the importance of PACAP in human conditions, because it’s therapeutic and biomarker potential have also been investigated in various human pathological conditions with encouraging results.

Author answer:

Thank you for your valuable suggestions. The authors in this section of the manuscript describe a more detailed protective role of the peptide (226-243).

  1. The manuscript would be of greater interest if the possible differences between the ASA vs. INDO vs. NA in MP, OSP and ISP should be also discussed as the changes in PACAP/SP/GAL-LI in MP vs. OSP vs. ISP.

Author answer:

Thank you for your comment. As recommended, the authors discussed and compared the results between ASA vs. INDO vs. NA in MP, OSP and ISP as well as changes in PACAP/SP/GAL-LI in MP vs. OSP vs. ISP (line 210-220).

  1. Lines: 259-260. The authors hypothesize, that NSAIDs might induce neuronal loss, resulting in an increase in the percentage of PACAP-, SP- and GAL-LI nerve cells. Was the presumed neuronal loss examined in this experiment?

Author answer:

Thank you for your valuable suggestions. The information contained in this sentence is only a hypothesis and speculation. Unfortunately, the authors did not conduct a study on neuronal loss.

MATERIALS AND METHODS

  1. This part of article would benefit from subsectioning f.e., animals, preparation of biological samples, IF method, statistical analyses.

Author answer:

Thank you for your comment. The authors corrected the text as recommended.

  1. Table 2 shows the reagents used in IF method. The PACAP antibody used in this experiment was PACAP1-27 antibody based on its code. The 38 amino acid residue (PACAP1-38, or PACAP38) is the predominant form in mammals, while PACAP27 represents only the ~10 % of total PACAP in the body. Why did the authors choose PACAP1-27 antibody instead of PACAP1-38?

Author answer:

Thank you for your comment. To date, both forms of PACAP have been studied within the gastrointestinal tract, although it is well known that PACAP-38 appears to predominate in mammalian tissues. The relatively low number of enteric neurons immunoreactive to PACAP-27 in the presented manuscript is in accordance with the previous studies, which established that the predominant form of PACAP in mammalian tissues is PACAP-38. Additionally, the biological effect of PACAP-27 and PACAP-38 is different which may be correlated to the fact that in numerous tissues PACAP-38  activates  intracellular signalling  mechanisms  that  differ  from  those  employed  by  PACAP-27. For example, in the rat small intestine PACAP-27 evoked potent secretory response not observed with PACAP-38 (doi: 10.1111/j.1476-5381.1992.tb14363.x.). Therefore, the determination of PACAP-27 expression in the jejunum seems to be relevant, interesting, and the choice of the form of the neuropeptide is correct. Certainly, this study will broaden the knowledge relating to the characteristics of the porcine enteric neurons in a physiological state as well as in drug-induced inflammation.

Round 2

Reviewer 1 Report

The manuscript has been sufficiently modified.

Reviewer 2 Report

I think this article was improved. The authours have been able to incorporate changes to reflect of the suggestions provided by me.